# Role of Interleukin-6 in Depressive Disorder

**DOI:** 10.3390/ijms21062194

**Published:** 2020-03-22

**Authors:** Emily Yi-Chih Ting, Albert C. Yang, Shih-Jen Tsai

**Affiliations:** 1Department of Psychiatry, Taipei Veterans General Hospital, Taipei 11217, Taiwan; et7334@gmail.com; 2Brain Research Center, National Yang-Ming University, Taipei 11221, Taiwan; accyang@gmail.com; 3Division of Interdisciplinary Medicine and Biotechnology, Beth Israel Deaconess, Medical Center, Boston, MA 02115, USA; 4School of Medicine, National Yang-Ming University, Taipei 11221, Taiwan

**Keywords:** interleukin-6, major depressive disorder, stress, polymorphism, antidepressant

## Abstract

Major depressive disorder (MDD), which is a leading psychiatric illness across the world, severely affects quality of life and causes an increased incidence of suicide. Evidence from animal as well as clinical studies have indicated that increased peripheral or central cytokine interleukin-6 (IL-6) levels play an important role in stress reaction and depressive disorder, especially physical disorders comorbid with depression. Increased release of IL-6 in MDD has been found to be a factor associated with MDD prognosis and therapeutic response, and may affect a wide range of depressive symptomatology. However, study results of the *IL6* genetic effects in MDD are controversial. Increased IL-6 activity may cause depression through activation of hypothalamic-pituitary-adrenal axis or influence of the neurotransmitter metabolism. The important role of neuroinflammation in MDD pathogenesis has created a new perspective that the combining of blood IL-6 and other depression-related cytokine levels may help to classify MDD biological subtypes, which may allow physicians to identify the optimal treatment for MDD patients. To modulate the IL-6 activity by IL-6-related agents, current antidepressive agents, herb medication, pre-/probiotics or non-pharmacological interventions may hold great promise for the MDD patients with inflammatory features.

## 1. Introduction

The etiology of major depressive disorder (MDD) is complex, with psychological, biological as well as environmental components. Cytokines, which are a heterogeneous group of polypeptides, may be the key players in the immune activation which has been repeatedly described in MDD and stress reactions [1]. Active transport systems may carry peripherally released cytokines from blood into the brain. Via a feedback mechanism, activated glia cells can produce cytokines [2]. Neurotransmitters, especially the monoaminergic systems that are thought to play a pivotal role in MDD are affected by the cytokines inside the brain [3]. As the production of the cytokines interleukin-6 (IL-6), tumor necrosis factor-alpha and interleukin-1 is associated with initiation of an inflammatory response, these cytokines are often referred to as pro-inflammatory cytokines. Implicated in the pathophysiology of MDD, these cytokines out of the activating cytokines are most relevant in terms of their actions on the brain [4,5,6].

Among these pro-inflammatory cytokines, animal studies and clinical studies had demonstrated that IL-6 may have a special role in the pathogenesis and somatic consequences of depressive disorder, as well as in the effects of depressive disorder treatment. In this review, we focus on the role of IL-6 in depressive disorder from various aspects. In this study, a systematic search was performed for IL-6 in depressive disorder by using the PubMed database up to September 1, 2019. Additional information from reference lists of published articles was also obtained. The following keywords were used for searching: depressi* [ti] AND ((interleukin-6) OR (IL6) OR (IL-6)). These literature searches were limited to English language articles. Additionally, manual searches for related articles were also performed.

## 2. Interleukin-6

Cloned and reported in 1986, the *IL6* gene encodes human IL-6 [7]. This gene consists of four introns and five exons, mapped to 7p15–p21 chromosome. With the 184 amino acid mature segment and a 28 amino acid signal sequence included, the *IL6* gene encodes a precursor protein with a total of 212 amino acids in length [7,8].

Proteins which use glycoprotein 130 (gp130; also known as CD130) to transduce signals include IL-6 as a part of the family. A receptor complex is formed by IL-6, which consists of a signal-transducing component gp130 and a ligand-binding IL-6 receptor (IL-6R). There are two types of IL-6 signaling: the anti-inflammatory pathway, or known as the classical way; the other is pro-inflammatory, the trans-signaling way [9]. IL-6 binds to the membrane bound IL-6R in the classical pathway. However, in the pro-inflammatory pathway, IL-6 attaches to a soluble IL-6R, which is not membrane-bound. Gp130 proteins attaches to both sides of this complex once it stabilizes on the membrane. The pro-inflammatory pathway (trans-signaling pathway) is used by various cell types within the brain. Signaling will not occur if gp130 proteins are soluble (not membrane-bound) acting as an antagonist, binding to the soluble IL-6R/IL-6 complexes [9].

Mature IL-6 is a pleiotropic cytokine with functions such as production of acute phase proteins in the liver, haematopoiesis, osteoclast activation, proliferation and differentiation of B lymphocytes and also the induction of fever in the brain [10]. IL-6 behaves both as a cytokine and myokine in the immune system, affecting many auto-immune diseases. IL-6 is well-known for its effects in diabetes, atherosclerosis, prostate cancer, encephalitis and also rheumatoid arthritis, acting as a pro-inflammatory cytokine reinforcing the inflammatory states [11].

## 3. IL-6 and Depression Animal Models

Inflammation, stress and depression are closely interrelated. Studies indicate that cytokine-mediated communication pathways between the immune system and the brain are involved in MDD pathogenesis [12]. Of the numerous cytokines, IL-6 is one of the most investigated cytokines in animal studies of MDD.

Murine restraint stress is commonly used in the study of behavioral and biological symptoms associated with MDD. Nukina et al. have tested whether or not the stress induced plasma IL-6 elevation in mice [13]. The plasma IL-6 concentrations increased after one hour of restraint stress and thereafter gradually decreased, suggesting that restraint stress is capable of elevating the plasma IL-6 levels [13]. Further study found that the induction of the elevated plasma IL-6 levels by restraint stress is independent of the intestinal microflora, the main source of the increase being the liver during stress [13]. An increase in *Il6* mRNA expression along with a four-fold increase in circulating IL-6 levels in the rat hypothalamus was found upon application of restraint stress [14]. A recent study using prolonged restraint stress in mice also found increased circulating *Il6* mRNA expression [15]. However, in that study, *Il6* mRNA was not increased in the brain during the stress or post-stress periods. A study done by Aniszewska et al. found that stress increased the number of IL-6-immunoreactive astrocytes and microglial cells while the levels of the IL-6R were increased in the hypothalamus [16].

In rodents, it has been widely reported that lipopolysaccharide (LPS, an endotoxin derived from gram-negative bacteria, which induces inflammation) administered either peripherally or centrally induce depressive-like behaviors and increases in cytokines, including IL-6 [17,18,19]. The study by Sukoff Rizzo et al. using intracerebroventricular administration of recombinant IL-6 also produced depressive-like behaviors in mice [20].

Studies indicated that *Il6* knockout mice showed less despair behaviors to stress. One of the studies carried out by Monje et al. indicated that *Il6* knockout mice became resistant to the development of depression-like symptoms following exposure to constant darkness (a chronobiologically induced depressive state paradigm), further supporting IL-6 having a functional role in the molecular mechanism of depression [21]. Likewise, Chourbaji et al. indicated that *Il6*-deficient mice (*Il6*(-/-)) exhibited reduced despair in the forced swim test and tail suspension test, and enhanced hedonic behaviors [22]. Another study found that the depressing effects of LPS intraperitoneal injection or injection into the lateral ventricle of the brain on depression-like behaviors were more marked in wild type than in *Il6*-deficient mice [23].

Ketamine has rapid and robust antidepressant effects in animal models and MDD patients. An animal study has investigated the antidepressant effects of ketamine and the IL-6 expression in the prefrontal cortex and hippocampus of a rat model [24]. The result demonstrated that, compared with the saline group, ketamine administration significantly decreased the immobility time of rats during the forced swim test and decreased the IL-6 expression in the prefrontal cortex and hippocampus, suggesting that ketamine-induced antidepressant effects are associated with decreased brain IL-6 levels.

A study tested the idea that with blocking of the IL-6R, such action may give a sustained and rapid antidepressant effect in animal model, as there is an increased finding of elevated IL-6 in MDD patients [25]. Rapid-onset and long-lasting antidepressant effects in susceptible mice was observed in this study when MR16-1, an anti-mouse IL-6R antibody, was intravenously injected, after social defeat stress through its anti-inflammatory actions [25]. On the other hand, MR16-1 intracerebroventricular injection in susceptible mice induced no antidepressant effects.

From the above studies, increased IL-6 levels whether by stress, LPS or direct IL-6 injection produced depressive-like behaviors in rodents. The IL-6 effect in depression is further supported by *Il6*-deficient mice, which were resistant to the development of a depression-like behaviors following exposure to stress.

## 4. Clinical Study of IL-6 in Depression

### 4.1. Stress, Depression and IL-6

In the development of depression, psychological stress plays an important role. Psychological stresses may be grouped into two classes in terms of the duration of stress: acute and chronic psychological stress, further classified as persistent or disconnected psychological stress [26]. In addition to activation of hypothalamic-pituitary-adrenal (HPA) axis and the sympathetic nervous system, increasing number of studies suggest that psychological stress has an important effect on elevated inflammation [27]. In a longitudinal community study assessing the relationship between chronic stress (caregiving for a spouse with dementia) and IL-6 production, it was found that caregivers′ average rate of increase in IL-6 was about four times as large as that of non-caregivers [28]. The link between stress and IL-6 was also demonstrated in a cross-sectional, school-based study with 370 adolescent boys [29]. It was found that adolescent life-event stress was a significant predictor of increased IL-6 in the study subjects. For acute stress such as operation, a study of ten patients undergoing hepatectomy found there were significant correlations between surgical stress and blood IL-6 levels [30].

Accumulating evidence from previous studies indicated an association between IL-6 cytokine and MDD, whether it be an increase or decrease in IL-6 levels in MDD patients [31]. Some other studies had found no changes in IL-6 when comparing MDD patients with controls. Three meta-analyses have verified that people with MDD show elevated serum/plasma IL-6 levels compared to people without depression [32,33,34]. It should be noted that several factors such as high body mass index (BMI), smoking and physical comorbidity may affect IL-6 levels, and people with depressive disorders frequently exhibit these lifestyle factors. Meta-analysis by Duivis et al. showed that, after controlling BMI, smoking and physical activity, the relationship between depression and IL-6 is no longer significant [35]. A total of 99 studies with different degrees of depression (MDD vs. depressive symptoms) were included in another meta-analysis [36]. In comparison to non-depressed groups, IL-6 was found to be elevated in the depressed groups in this meta-analysis. Such elevation was more obvious in subgroups where MDDs were compared to those with only depression asserted by standard inventory in terms of diagnosis. The same goes for subjects who were selected for cardiovascular disease compared to ones not selected for particular comorbidity [36]. In studies where the MDD inflammatory hypothesis is tested, comorbidity and depression severity need to be measured and controlled.

Previous studies of serum IL-6 levels and MDD are mostly cross-sectional designs. Zalli et al. have carried out a study which investigated on the temporal relationship between serum IL-6 levels and the persistence of depressive symptoms among older participants [37]. Their results indicated that no cross-sectional association was found between IL-6 levels and depressive symptoms at baseline. However, higher IL-6 levels predicted depressive symptoms at 5-year follow-up that is independent of age, gender, BMI, smoking status or cognitive function [37]. Another longitudinal study in approximately 4500 individuals demonstrated that higher serum levels of IL-6 in childhood (age 9) would increase future risks for depression at age 18 [38]. However, in another study with 4756 women, blood levels of IL-6 were not associated with incident depression over a follow-up of 6-18 years [39].

Study by Engler et al. directly tested the intravenous administration of low-dose endotoxin, one of the most important bacterial components contributing to the inflammatory process, in healthy male volunteers [40]. They found selective increase of IL-6 in the cerebrospinal fluid (CSF) and a strong association between the increase of CSF IL-6 levels and the endotoxin-induced mood deterioration, suggesting that increase in the central IL-6 concentrations may contribute to depression pathogenesis.

### 4.2. IL-6 and Depression Subtypes

Depression is a complex psychiatric condition with different subtypes. Studies have investigated peripheral IL-6 levels in different depression subtypes or their correlations with certain depressive symptoms.

Melancholic depression, a relatively homogenous MDD subtype, is typified by features of psychomotor retardation, anxiety, appetite loss and sleep changes. In 1993, Mae et al. have first demonstrated that IL-6 activity was significantly higher in melancholic depression than in healthy control subjects, and in patients with minor depression or non-melancholic depression [41]. Melancholic depressed patients had significantly higher baseline IL-6 levels that did not normalize despite clinical response to electroconvulsive therapy (ECT) in a study done by Rush et al., when compared with normal controls [42]. A recent systematic review of eight studies also showed that serum IL-6 values increased in melancholic depression group in comparison to controls and non-melancholic depression groups, confirming the findings previously mentioned [43].

Atypical depression, which is referred to as MDD with atypical features, is actually quite common and around 15% to 29% of MDD patients have atypical depression [44]. Unlike other forms of MDD, people with atypical depression may have hypersomnia, increased appetite and weight gain. A study with patients with typical and atypical MDD showed that, compared to normal controls, IL-6 levels were elevated in atypical MDD patients but not in patients with typical MDD [45].

Dysthymic disorder is a disorder phenotypically similar to but less severe than MDD. A study in patients with dysthymic disorder and MDD found that the plasma IL-6 levels in both groups were significantly higher than those in the control group [46]. When comparing the major depression group and the dysthymic group, no difference was observed in the plasma IL-6 levels.

Postpartum depression (PPD) is a subtype of depression associated with childbirth. In a study of 296 women with childbirth [47], 45 (15.2%) were found to meet the criteria for PPD during the follow-up. Serum IL-6 levels in women with PPD were significantly higher than those without PPD. Another study obtained serum sample at the time of delivery and found IL-6 levels at delivery were not related to depression during the first six months after delivery [48].

Inflammation may play a pathological role in psychiatric disorders especially in the elderly patients because a pro-inflammatory state is associated with ageing. Several studies have investigated whether peripheral IL-6 levels are significantly higher in the elderly with depression. Ng et al. carried out a meta-analysis of nine studies to evaluate whether peripheral IL-6 levels are significantly higher in elderly with depression [49]. They found, compared with controls, IL-6 were significantly elevated in the aged depression subjects. Similarly, in a study with 86 dementia-free women aged 70–84 years, women with current depression had higher levels of CSF IL-6 compared with those without depression [50].

Exposure to adverse childhood experiences (ACE) is associated with increased risk of development of MDD later in life. A study has demonstrated that MDD patients with ACE, but not MDD patients without ACE, showed significantly higher IL-6 concentrations compared to healthy controls [51].

Study has tested the association between IL-6 levels and depression severity. Recent study by Fan et al. indicated that IL-6 levels are positively associated with Hamilton Depression Scale-17 scores for MDD patients [52]. In another study, results indicated that IL-6 serum levels are elevated during the acute state of depressive episodes compared with controls, and that, in remission, the levels of IL-6 in depressed patients are consistent with those of controls [53].

### 4.3. IL-6 and MDD Features

IL-6 levels may be associated with the phenotypes of MDD. An interesting research done by Kakeda et al. investigated the relationship between IL-6 levels and brain morphology in first-episode drug-naïve MDD patients [54]. They found that the prefrontal cortex thickness was significantly reduced in MDD patients, and showed a significant inverse correlation with the serum IL-6 level. High levels of serum IL-6 were associated with reduced left subiculum and right CA1, CA3, CA4, GC-DG, subiculum and whole hippocampus volumes in MDD patients [54].

Concentration difficulties were often noted in MDD patients that may affect their day-to-day function. A study of attention in patients with MDD found that increased IL-6 levels were associated with the impairment of sustained attention [55]. In an earlier study done by Gimeno et al., they found that IL-6 levels predicted cognitive symptoms of depression in an average follow-up of 11.8 years, while baseline symptoms of depression did not predict IL-6 levels at follow-up, suggesting that inflammation precedes depression at least with regard to the cognitive symptoms of depression [56].

Insomnia is commonly noted in patients with MDD. A recent study of 44 MDD patients showed that sleep disturbances in MDD were positively correlated with plasma IL-6 levels [57].

### 4.4. IL-6 in Physical Diseases Associated with Depression

Many physical diseases are comorbid with depression, mainly cancer and inflammatory diseases associated with higher risks of developing depression [58,59]. Studies had tested the association of IL-6 levels in physical disorders associated with depression. Most studies found that patients comorbid with depression have elevated IL-6 levels than those without depression (Table 1). 

In 2001, Musselman et al. first reported that cancer patients with depression had significantly higher plasma IL-6 levels than healthy control subjects and cancer patients without depression [60]. Similar findings were demonstrated that the depression state was positively correlated with IL-6 levels in patients with cancer [61], lung cancer [62], colorectal cancer [63], pancreatic cancer [64], breast cancer [65,66], metastatic cancer [67], advanced cancer [68] and ovarian cancer [69]. One study investigated IL-6 level changes in six months following primary treatment in ovarian cancer patients [70]. It was found that improvement in vegetative depression is associated with the normalization of IL-6 levels. From the above consistent findings, cancer patients may be especially vulnerable to the prodepressive effects of IL-6 [11]. One study investigated 112 terminally ill cancer patients who only accepted palliative care [71]. The study found that the vegetative symptoms, such as appetite loss, insomnia and fatigue, but not depression symptoms, were significantly associated with IL-6 levels.

Depression is a common comorbidity in patients with cardiovascular disease. In a study of out-patients presenting with cardiovascular risk factors, elevated IL-6 serum concentration was linked to the presence of depressive mood [72].

In a recent study of 120 patients with rheumatoid arthritis (RA), symptoms of depression were found in 91 patients [73]. No significant differences were found between the values of depression ratings and the IL-6 levels. However, a study in Taiwan with 113 RA patients and 42 healthy controls demonstrated elevated serum IL-6 and IL-17 levels in RA patients with depressive symptoms [74].

Report indicated that patients on maintenance hemodialysis (HD) with symptoms of depression had higher serum IL-6 levels [75]. This finding is replicated by several studies [76,77,78]. Similarly, findings from a recent report also suggested that circulating IL-6 can be used as a biomarker for prediction of depressive symptoms in HD patients [79]. Another study found a correlation of IL-6 levels with both depression and fatigue in HD patients [80]. However, another study found that the relationship between depression and IL-6 levels were observed in patients on continuous ambulatory peritoneal dialysis, but not in HD patients [81].

In a cross-sectional study investigating the prevalence of depressive state in knee osteoarthritis (OA) patients, 52% patients were in the depressive state [82]. The study found that the serum IL-6 levels are associated with depressive state in the OA patients irrespective of disease severity.

Depression is commonly found in multiple sclerosis during relapses. A study found that the IL-6 levels increased during the acute phase of the disease, especially when depression is detected [83].

Studies have investigated the association between peripheral IL-6 levels and depression in patients who accepted medical treatment. Interferon-alpha therapy may result in depression in some people. In a cohort of 95 non-depressed hepatitis C patients followed for four months during interferon-alpha therapy, higher pre-treatment IL-6 levels were associated with the following incidence of depression [84]. A study has evaluated the relationship between depression and IL-6 and IL-10 in patients undergoing hematopoietic stem cell transplantation. The result showed that patients with depression had significantly higher levels of serum IL-6 and IL-6-to-IL-10 ratios compared to patients without depression [85]. In fifty patients undergoing coronary artery bypass grafting procedures, depressive symptom severity was measured at baseline, discharged, with a six-month follow-up [86]. Results demonstrated that changes in depression severity were associated with incremental changes in blood levels of S100B, but not IL-6. In 110 patients undergoing total knee replacement, it was found that in-hospital IL-6 levels predicted depressive symptoms at three months following surgery [87].

### 4.5. IL6 Genetic Studies in MDD

The etiology of MDD has not been clearly identified, although evidence supports neurobiological and genetic origins. The possibility that *IL6* mutations contribute to MDD susceptibility has been investigated in many genetic studies, but the results were inconsistent.

Among the *IL6* polymorphisms, the -634C>G (rs1800796) polymorphism was the most commonly studied. The *IL6* (-634) promoter mutation affects *IL6* transcription and expression directly. Analysis of the IL-6 secretion data reveals that individuals carrying the -634G allele have a higher IL-6 secretion capacity than those with the -634CC homozygotes [88]. In 2005, we first investigated whether the *IL6* -634C>G genetic variant confers susceptibility to MDD [89]. Our findings suggested that the investigated *IL6* polymorphism does not affect MDD susceptibility.

A G/C transversion 174bp upstream of the transcription start site of the *IL6* (rs1800795) gene is another functional single nucleotide polymorphism (SNP). Promoters with the G allele at this site showed high affinity binding to GATA binding protein-1 (GATA1) transcription factor in the region spanning -177/-168bp [90]. The study done by Fishman et al. also demonstrated that the G allele of this SNP was associated with higher plasma IL-6 concentrations when compared to that of the C allele carriers in patients with systemic-onset juvenile chronic arthritis [91]. These results imply that this rs1800795 polymorphism may provide a molecular biomarker for MDD. In a study including men and women aged 70-80 years, increased 10-year mortality risk associated with late-life depressive symptoms occurred for individuals who carry the homozygous GATA1-sensitive *IL6* -174G allele, yet this was not found in those homozygous for -174C allele or heterozygotes [90]. A study in a Hungarian population sample of 1053 volunteers found that this functional *IL6* rs1800795 polymorphism interacted with the recent negative life events to increase the risk of depression [92]. However, the CC genotype carriers achieved significantly higher depression scores than that of -174G allele carriers. Similarly, another study in 444 Australian youths showed that the *IL6* -174G allele carriers had fewer depressive symptoms following interpersonal stress, but not for other stressors, relative to -174CC homozygotes [93]. They also indicated that -174G confers protection against inflammation in adolescence, yet it also increases the risk for inflammation in adulthood, suggesting an age-dependent factor. Another depression genetic study tested 11 cytokine genetic variants, including *IL6* rs1800795, and all the tested SNPs were not associated with early-onset mood disorders [94]. Single genes may offer little predictive power for the identification of MDD. For example, Roetker et al. found that women who were homozygotes (-174CC or -174GG) for rs1800795 are not associated with depression but had increased risk of depression, under the condition that other risk genes were also present [95].

A study done by Zhang et al. investigated various *IL6* SNPs (rs1800797, rs1800796, rs1800795, rs2069837 and rs1524107) in MDD [96]. They demonstrated that there is a difference in the allele frequencies of *IL6* rs1800797 SNP between MDD patients and controls [96]. Furthermore, the expression quantitative trait loci analysis showed a marginally significant association between the rs1800797 and *IL6* expression in the frontal cortex. In a large genome-wide association analysis for MDD, 44 independent and significant loci were identified but *IL6* variants were not among the hits [97].

DNA methylation has been regarded as a potential link between environment and depression. One research investigating *IL6* methylation in late-life depression showed subjects with depression had lower *IL6* methylation levels at one of the four sites investigated [98]. There was no effect modification when considering *IL6* genotype.

IL-6 exerts its biological activities through two molecules: IL-6R and gp130. A study done by Khandaker et al. indicated that a common functional variant in the *IL6R* gene (*IL6R* Asp358Ala; rs2228145 A > C), which is able to decrease inflammation by impairing IL-6R signaling, is associated with depression risk [99]. This result further confirms that IL-6 signaling pathway may be implicated in MDD.

Among the common functional *IL6* genetic variants, some studies demonstrated that the *IL6* rs1800795 C allele increased the risk for depression, however not by all [90,92,93,94,96]. Investigating a single *IL6* polymorphism might result in overlooking of some information from the other *IL6* SNPs. Through the reduction in number of polymorphisms to be genotyped by using a haplotype constructed from several tag SNPs, genotyping efficiency can be improved. Other genetic variants affecting the gene function may also be tagged by this haplotype.

The *IL6* genetic effects on the MDD susceptibility be affected by environmental factors such as stress in early childhood or physical diseases. There is a chance to identify the *IL6* genetic effects on the MDD development with the emergence of machine learning and artificial intelligence for future research in gene-environment interactions and epigenetics [100,101].

The above clinical studies suggested that higher blood IL-6 levels are associated with MDD or depression that manifests in patients with a physical disorder. IL-6 levels are also related to the development, manifestations and severity of MDD. However, evidence for the involvement of *IL6* in the MDD susceptibility is currently inconsistent.

## 5. IL-6 and Depression Treatment

Several modalities have evidence for efficacy in MDD treatment. Among them, antidepressants are the most commonly used to treat depression. Other strategies include exercise, electroconvulsive therapy (ECT), psychotherapy in its many forms and phototherapy. Studies have tested the effect of depression treatments on peripheral IL-6 levels.

### 5.1. Medications

In a study with 51 MDD patients, plasma IL-6 levels were elevated during the acute state of MDD compared to controls [102]. Treatment with selective serotonin reuptake inhibitor (SSRI) antidepressants significantly reduced plasma IL-6 levels. In addition, this study found that the plasma IL-6 level was higher in antidepressant-resistant than antidepressant-responsive MDD patients [102]. In another study with 118 MDD patients treated with SSRI antidepressants, in comparison to SSRI non-responders, the IL-6 baseline levels in SSRI responders were significantly higher, indicating a significant correlation between plasma IL-6 levels and changes in severity of depressive state [103]. These findings implicated that baseline plasma IL-6 levels might be a biological marker for antidepressant treatment response. These clinical findings are in line with the rodent depression model that showed SSRI (fluoxetine) treatment reversed isolation-induced depressive-like behaviors and suppressed cytosolic IL-6 protein [104]. It should be noted that the early study by Maes et al. did not find any effects of tricyclic antidepressants on serum IL-6 levels in MDD patients [105]. This implied that different antidepressants may have various effects on IL-6 levels.

IL-6 was found to be elevated in the depressed population, and serum IL-6 levels in depressed patients decreased at 4 h after ketamine infusion, but then returned to baseline 24h after infusion, as indicated by a recent clinical study [106]. A small scale ketamine treatment in an MDD study found that baseline serum IL-6 levels could be a useful predictive biomarker for ketamine treatment in treatment-resistant depression patients [107].

Since the MDD pathophysiology is associated with the hyperactivity of inflammatory responses, clinical trials have tested the antidepressant effects of antiinflammatory drugs like celecoxib. In a placebo-controlled study of 40 patients with MDD, the adjunctive celecoxib group showed better response rates than the placebo group [108]. In this study, significant correlation was found between depression severity reduction and decrease of serum IL-6 levels at week 6, suggesting that the antidepressant effect of celecoxib might be linked to its capability of decreasing IL-6 [108].

Statins are a class of lipid-lowering agents and are thought to possess antiinflammatory properties. A study in 222 stroke patients followed up for one year found the preventive effects of statin use against post-stroke depression [109]. Furthermore, the study found that there was a significant interaction between statin use and IL-6 on the presence of a depressive disorder at the 1st year.

In humans, the effects of the IL-6 antibodies (sirukumab and siltuximab) on depressive symptoms have been investigated in 176 RA patients and 65 multicentric Castleman′s disease patients [110]. Analysis showed that, compared with placebo, sirukumab and siltuximab made significantly greater improvements on depressive symptoms in these patients [110].

### 5.2. Natural Products

Some natural products have been tested for their antidepression effects as well as effects on the IL-6 levels. A study done by Shukkoor et al. investigated the antidepressive effects of lipid extract from *Channa striatus*, which is a freshwater fish in Malaysia [111]. The extract gave significant antidepressant-like effects in the chronic unpredictable mild stress model in rats and also reversed the IL-6 levels elevated by stress.

Baicalin, one of the main active ingredients of the traditional Chinese medicine drug Scutellaria radix, was investigated for its antidepressive effects in mice with chronic unpredictable mild stress [112]. Baicalin treatment showed antidepressant effects and considerably inhibited proinflammatory cytokine levels, including IL-6.

Chai-hu-gui-zhi-gan-jiang-tang, a Chinese herbal medicine, has been studied in peri- and post-menopausal women with depression [113]. The study found that both soluble IL-6R and plasma IL-6 concentrations were reduced by Chai-hu-gui-zhi-gan-jiang-tang in relation to the improvements in depressed mood during treatment.

### 5.3. Exercise

Apart from using medications as treatment for MDD patients, other studies had investigated the impacts of exercise on peripheral IL-6 levels. As indicated by research studies, vigorous exercise in patients increased serum IL-6 levels, yet moderate or low levels of exercise lowered IL-6, with preference for moderate level as being the most effective [114]. Another highlight of this study is that there is a stronger association between subjects with higher serum IL-6 baseline levels and their IL-6 reductions after interventions. In another study, MDD patients who were partial responders to an SSRI were randomized to receive one of two doses of exercise [115]. They found a significant positive correlation between change in blood IL-1beta and change in depression symptom scores. There were no significant changes in mean level of any cytokine, including IL-6, following the exercise intervention [115].

Depression is commonly found in patients with chronic obstructive pulmonary disease. A study investigated the effect of aerobic exercise in chronic obstructive pulmonary disease patients [116]. They found aerobic exercise significantly decreased depression and peripheral inflammation markers including IL-6 in these patients.

Hemodialysis patients have a higher risk of depression. The non-pharmacological strategy, bicycle riding, has been shown to ameliorate the depression severity of the patients undergoing hemodialysis by decreasing the levels of IL-6 and IL-18 [117].

Yoga showed benefits as an adjunctive intervention for MDD subjects who have inadequate response to antidepressant treatment and a significant reduction in IL-6 levels were noted after the yoga treatment [118].

### 5.4. Electroconvulsive Therapy

Among the treatments for MDD patients, ECT occupies an important place, especially for patients with antidepressant-resistance or having strong suicide risks. A study by Järventausta et al. have tested the acute and long-term changes in IL-6 levels after ECT [119]. They found that ECT has distinct acute and long-term effects on IL-6 levels that an increase in IL-6 levels soon after ECT intervention. However, the baseline IL-6 levels decreased among remitters after repeated ECT treatment. The findings are replicated by the following study and further demonstrated that IL-6 levels prior to ECT treatment may be a useful marker in identifying those MDD patients most likely to respond to ECT treatment [120].

Repetitive transcranial magnetic stimulation (rTMS) has become an alternative to ECT for treatment of severe depression. So far, there is no study about the rTMS effect in IL-6 levels which may warrant further study.

### 5.5. Light Therapy

Light therapy is a way to treat MDD, particularly seasonal affective disorder (SAD), a subtype of MDD. In a study of 15 patients with seasonal affective disorder, these patients had significantly increased IL-6 levels compared to the normal controls [121]. Although light therapy improved depression severity, IL-6 levels were not altered by two weeks of successful light therapy.

A direct relationship between sunlight exposure and plasma IL-6 level changes in a study where the effects of sunlight exposure on plasma IL-6 in depressive and non-depressive subjects were investigated [122]. IL-6 levels were not affected by exposure to various different intensities of light in non-depressed subjects. Levels of IL-6 were lower in the low light exposure depressive group compared to the high light exposure to depressive groups. The hypothesis that impairment of the suprachiasmatic nucleus leads to different IL-6 levels in the depressive subjects was established in this study, and that MDD could benefit from sunlight exposure [122].

### 5.6. Psychological Interventions

Psychological interventions (such as cognitive behavior therapy, mindfulness-based cognitive therapy, cognitive behavioral analysis system of psychotherapy and interpersonal psychotherapy) is an effective treatment for MDD. In a randomized clinical trial in MDD patients, results showed that serum IL-6 levels, as well as depression severity, significantly decreased after 16 sessions of supportive-expressive dynamic psychotherapy [123]. It was found that after individual mindfulness training, the salivary IL-6 levels had decreased, which was maintained for 3 months, in a study testing the application of brief mindfulness intervention in young women with depressive symptomatology [124]. One small scale study with 11 women in the first episode of depression treated with cognitive behavioral therapy alone demonstrated that cognitive behavioral therapy reduced both depressive symptoms and the serum IL-6 levels [125].

As shown above, there are a number of pharmacological agents and non-medication treatments for MDD that have shown effect on the IL-6 levels. IL-6 may represent a valid and highly effective target for the development of novel antidepressant treatments.

## 6. IL-6 and the MDD Pathogenesis

Major depression is a heterogeneous disorder, which has many causes and may not be explained by a single mechanism. There are many hypotheses proposed for depression pathogenesis, including neurotrophic hypothesis, monoamine hypothesis, neuroinflammatory hypothesis, dysfunctional HPA axis hypothesis, etc. From the biopsychosocial model, there are two fundamental etiological perspectives about MDD; biomedical factors (e.g., genetic vulnerability, physical diseases and environmental pollutant) and psychosocial stress.

Figure 1 presents the possible role of IL-6 in MDD pathogenesis. Here, when encountering molecular pathogens or foreign antigen, monocytic immune cells respond by secreting cytokines including IL-6 [126]. Dendritic cells mature following antigen presentation, cytokine production and co-stimulation from the system, which in turn activates lymphocytes, such as T cells [126].

The blood brain barrier (BBB) does not allow the relatively large cytokine molecules to freely pass through. However, as evidence suggests that neural, cellular and humoral pathways are used for cytokine signals to reach the brain [127]. The permeability of BBB may be potentially increased by circulating inflammatory factors, allowing access to the brain by pro-inflammatory cytokines released by activated macrophages and monocytes, such as IL-6 via the leaky regions of the BBB, including the circumventricular organs and the choroid plexus [127,128]. Moreover, there may be compromise of the intestinal barrier functions by activation of the inflammatory response system via an increase in IL-6 production [129]. Loss of function of the protective barrier may be caused by the latter, leading to the increase in the gram-negative bacteria translocation (leaky gut) [129].

Both clinical and animal studies also indicated that stress is sufficient for increases in IL-6 levels [130,131]. Elevated IL-6 might then cause HPA axis dysfunction, alterations in synaptic neurotransmission and reducing neurotrophic factors (Figure 1).

One’s capacity to deal with stress is mainly regulated by the HPA axis. The HPA-axis is able to influence a wide range of physiological processes such as digestion, immune response, emotions, energy metabolism and sexual behaviors. Under normal conditions, the HPA axis self-regulates and its activity is managed by a negative feedback inhibition. The glucocorticoid receptor (GR) binding in the hypothalamus and the pituitary blocks HPA axis activity and downstream release of glucocorticoids from adrenal cortex. Psychosocial stress, by initiating changes in the HPA-axis and the immune/inflammatory system, acts as a trigger for depression development. Depression associated with hypercortisolemia showing increased activity of HPA-axis, and inhibitory feedback reduction are the most consistent findings in literature [132]. On the other hand, stress-induced elevating IL-6 signaling in the hypothalamus has also a modulatory effect on HPA-axis activation [133,134].

There is a well-established link between pro-inflammatory cytokines and decreases in monoamine synthesis [132]. Studies had indicated that IL-6 may increase the activity of indoleamine-2,3-dioxygenase (IDO), which is responsible for the catalysis of tryptophan, leading to kynurenine pathway activation and decreasing central serotonin availability [135]. This results in the synthesis of the neurotoxic N-methyl-d-aspartate glutamate agonist quinolinic acid and 3-hydroxykynurenine thereby enhancing oxidative stress and contributes to neurodegeneration which characterizes MDD particularly in late life [136].

The evidence addressing the interaction between cytokines and neurotrophins in the brain is scarce, despite the fact that there is accumulating evidence showing involvement of neuroinflammation and brain-derived neurotrophic factor (BDNF) in brain disorders [137]. NF-κB-dependent pro-inflammatory activation of microglia is induced in chronically stressful situations. A decrease in the BDNF signaling in the synaptic cleft results from the binding of pro-inflammatory cytokines to microglial cells [137]. Considering the important role of IL-6 and BDNF in MDD pathogenesis, further study is needed to test the interaction between the two factors.

## 7. Perspectives and Future Directions

Many findings support the involvement of IL-6 in the pathophysiology of MDD but some findings are against (Table 2). On the one hand, increased IL-6 levels are associated with both human depression and a range of rodent models of the disorder. Genetic knockout of *Il6* demonstrate less despair behaviors to stress. On the other hand, a number of genetic association studies of IL6 in MDD have generated negative results. A number of clinically effective antidepressants decrease IL-6 levels, while tricyclic antidepressants have no effect on serum IL-6 levels in MDD patients.

### 7.1. Could Blood IL-6 Levels be Used as Biological Marker for Depression Diagnosis and Treatment?

Biomarkers help in diagnostic accuracy and provide a potential target for identifying predictors of response to various interventions. So far, there are still no biomarkers for MDD that could be introduced into clinical practice. There are numerous reports which demonstrated the association between elevated peripheral IL-6 levels and depression, especially in physical diseases associated with depression (Table 1). IL-6 levels have also been linked to MDD features, subtypes and therapeutic response.

However, a biomarker for clinical use needs good sensitivity and specificity. In addition to IL-6, some other cytokines or their receptors (such as tumor necrosis factor-alpha, interleukin-1 beta and IL-6R) may act synergistically on MDD pathogenesis. A single particular cytokine, like IL-6 is far from ideal in guiding diagnosis and therapeutic intervention, but a combination of multiple cytokines may improve the accuracy. For example, a study by Xu et al. developed a cytokine scoring system, which integrated the four cytokines into one score system that performs well in disease severity and fatality prediction in pediatric septic shock patients [138]. Future studies are needed to integrate the depression-related cytokine levels to classify biological subtype to advance MDD precision medicine interventions.

### 7.2. How to Modulate IL-6 Activity to Treat Depression?

The high IL-6 level found in MDD patients in many clinical studies is even more apparent in MDD patients who are treatment-resistant. As with patients who responded to treatment in particular, there is an association between the decrease in the peripheral IL-6 levels and the treatment itself. Therefore, in order to proceed with clinical recovery for depression, it may be concluded that suppression of IL-6 activity is required. Using IL-6R antibodies (e.g., tocilizumab) or IL-6 antibodies (e.g., sirukumab) seems to be a direct strategy to reduce the IL-6 activity. Some initial animal and clinical studies indicated potential antidepressive effect of IL-6 antibodies [25,110]. Clinical observation also indicated potential benefit of sirukumab, an anti-IL-6 monoclonal antibody, for RA patients with depression [110]. However, the cost, the BBB permeability and adverse effects may limit their use in MDD treatment. Furthermore, there are two types of IL-6 signaling: a classical anti-inflammatory signaling and a trans-signaling proinflammatory signaling. Treatment target in depression such as IL-6R inhibitors or IL-6 may not be the optimal choices [139]. Maintenance of the classical IL-6 signaling with an increase in the soluble gp130 inhibition in IL-6 trans-signaling may be a more promising target [139].

Earlier studies have found that antidepressants, exercise, some natural products, light therapy, ECT and psychological interventions may normalize IL-6 levels. However, the underlying mechanisms for such effect need further exploration. Furthermore, not all antidepressants showed a lowering IL-6 level effect. Identifying current antidepressants with more IL-6 normalization effects may help to treat those MDD patients with elevated IL-6 levels.

### 7.3. How to Reduce IL-6 Activity through Gut Microbiota Modulation?

The pathophysiology of depression is multifactorial. Studies in recent decade has pointed out that stress-induced changes in the gut microbiota would lead to a systemic inflammation which in different ways would modulate brain inflammatory pathways and cause depression [140]. Some preclinical studies had demonstrated that probiotics, which have potential antidepressant properties, can attenuate stress-inducing IL-6 [141,142]. Clinical confirmation of these findings might offer a useful novel therapeutic approach to normalize IL-6 levels in MDD.

## 8. Conclusions

With the important role of inflammatory processes in depression pathogenesis, among the cytokines, IL-6 has been consistently found to be elevated in stress reactions and depression patients. IL-6 levels have been associated with MDD subtypes, clinical features, therapeutic responses and prognosis. All this evidence strongly supports that restoring IL-6 activity is the key to treating depression involving inflammation. In addition, the combination of IL-6 and other cytokine levels may help to group MDD into biological subtypes for precision medicine. While the need for novel antidepression therapies remains high, it is imperative that a greater understanding of the complex and apparently dichotomous effects of IL-6 manipulation in the brain is first acquired.

## Figures and Tables

**Figure 1 ijms-21-02194-f001:**
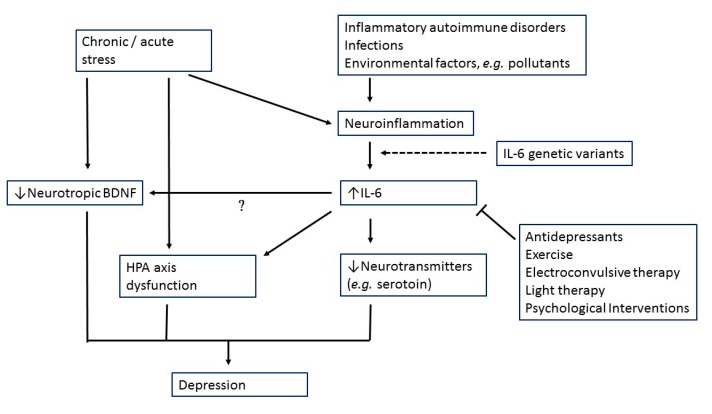
Possible role of interleukin-6 (IL-6) in major depression pathogenesis.

**Table 1 ijms-21-02194-t001:** Studies of IL-6 levels in physical diseases associated with depression.

Physical Disorders	IL-6 Levels in Patients with Depression	Reference
Cancer	Higher plasma IL-6 levels in patients with depression.	[60]
Cancer	Depression state was positively correlated with IL-6 levels.	[61]
Lung cancer	Depression status was positively correlated with IL-6 levels.	[62]
Colorectal cancer	Serum IL-6 levels revealed positive associations of depression.	[63]
Pancreatic cancer	An association between depression and serum IL-6.	[64]
Breast cancer	Depression group had markedly higher plasma IL-6 levels than the other group.	[66]
Breast cancer	Plasma IL-6 was significantly correlated with symptoms of depression.	[65]
Breast cancer	Plasma IL-6 is strongly associated with depression.	[67]
Advanced cancer	Among those whose blood was drawn within 48 h of interview completion, depression and plasma IL-6 were highly correlated.	[68]
Ovarian cancer	Greater vegetative depression was related to elevated plasma IL-6	[69]
Terminally ill cancer patients	Neither of the depressive symptoms nor their severity was associated with plasma IL-6 levels.	[71]
Cardiovascular risk factors	Elevated serum IL-6 levels are linked to the presence of depression.	[72]
Rheumatoid arthritis	No significant differences were found between depression scale and serum IL-6 levels	[73]
Rheumatoid arthritis	There was a direct correlation between depression rating and IL-6 levels.	[75]
Hemodialysis	Serum IL-6 levels were positively correlated with the values of depression scale.	[78]
Hemodialysis	Depressed patients had higher serum IL-6 levels.	[77]
Hemodialysis	Depressed patients showed an increase serum IL-6 levels.	[76]
Hemodialysis	Serum IL-6 was higher in patients with depressive symptoms.	[79]
Hemodialysis	A correlation of IL-6 levels was found with both depression and fatigue in hemodialysis patients.	[80]
Hemodialysis	No relationship was found between depression and plasma IL-6 levels.	[81]
Continuous ambulatory peritoneal dialysis	In patients without depression, plasma IL-6 levels were significantly lower.	[81]
Osteoarthritis	The serum IL-6 levels were associated with depression severity.	[82]
Multiple sclerosis	Increase in interleukin-6 levels is related to depressive symptoms.	[83]
Interferon-alpha therapy	High circulating IL-6 levels may be risk factors for interferon-alpha induced depression.	[84]
Hematopoietic stem cell transplantation	Patients with depression showed significantly higher serum IL-6 levels.	[85]
Coronary artery bypass grafting	Acute changes in depressive symptom severity were not associated with IL-6 levels.	[86]
Total knee replacement	Plasma IL-6 levels predicted depressive symptoms at three-months following surgery.	[87]

**Table 2 ijms-21-02194-t002:** Evidence for and against the role of IL-6 in depression.

**Evidence for**
1. Preclinical studies suggest that increased IL-6 levels whether by stress, lipopolysaccharide or direct IL-6 injection produced depressive-like behaviors in rodents. 2. *Il6* knockout mice showed less despair behaviors to stress. 3. Most published findings on serum/plasma IL-6 have demonstrated a correlation between high levels of this interleukin and MDD. 4. Longitudinal study demonstrated that higher serum IL-6 levels would increase future risks for depression. 5. IL-6 antibodies made significantly greater improvements on depressive symptoms in patients with autoimmune diseases.
**Evidence against**
1. Results of *IL6* genetic association studies in MDD have been inconsistent. 2. Different types of antidepressant medication may have divergent effects on IL-6 levels. 3. Intracerebroventricular anti-IL-6R antibody injection in susceptible mice induced no antidepressant effects.

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
