# Peer review of "Role of Interleukin-6 in Depressive Disorder"

_ijms, 2020, doi:10.3390/ijms21062194_

Round 1

Reviewer 1 Report

The manuscript of Ting et al. provides a comprehensive summary of the role of Interleukin-6 in depression. While the breadth of the review is impressive, I‘d suggest a few improvements:

Major

- It remains unclear whether the authors used systematic search criteria (e.g. which literature databases [when?], which search terms) to find the literature cited. This should be mentioned.

- On p. 9, the authors mention the atypical depression „is actually quite common“. This should be further specified and backed up by literature references.

- Dythymia (p. 9) is, AFAIK, not a type of depression but a disorder phenotypically similar to but less severe than depression.

- There review is very comprehensive, which can leave readers a bit overwhelmed. The manuscript would benefit from a succinct summary sentence at the end of each section, and maybe an additional textbox/table/graphic the provides a short overall summary.

- On p. 21, psychological interventions are pictured as additional treatments that are used only if medication is not effective. This is misleading, as the treatment of depression always relies on many different interventions in parallel.

- There are two secions on genetics (p. 13 and p. 24). For consistency, these should be combined. Importantly, the authors should also briefly mention whether IL-6 variants are among the hits in large GWAS for depression (e.g. PMID:29700475).

- Animal studies of antidepressant medication are now included with human studies. For consistency, there should be a separate section (such as section 3.).

Minor

- The manuscript would benefit from being proof-read by a native English-speaker.

Author Response

The manuscript of Ting et al. provides a comprehensive summary of the role of Interleukin-6 in depression. While the breadth of the review is impressive, I‘d suggest a few improvements:

Response: Thank you for reviewing this work and for the constructive feedback.

Major

- It remains unclear whether the authors used systematic search criteria (e.g. which literature databases [when?], which search terms) to find the literature cited. This should be mentioned.

Response: Thank you for pointing out this. We have added the search criteria to find the literature cited.

“In this study, a systematic search was performed for IL-6 in depressive disorder by using the PubMed database up to September 1, 2019. Additional information from reference lists of published articles was also obtained. The following keywords were used for searching: depressi* [ti] AND ((interleukin-6) OR (IL6) OR (IL-6)). These literature searches were limited to English language articles. Additionally, manual searches for related articles were also performed.” (Page 3; Line 19)

- On p. 9, the authors mention the atypical depression „is actually quite common“. This should be further specified and backed up by literature references.

Response: We have specified this and added the literature reference.

“Atypical depression, which is referred to as MDD with atypical features, is actually quite common and around 15% to 29% of MDD patients have atypical depression [44].” (Page 10; Line 5)

- Dythymia (p. 9) is, AFAIK, not a type of depression but a disorder phenotypically similar to but less severe than depression.

Response: We revised the description to “Dysthymic disorder is a disorder phenotypically similar to but less severe than MDD.” (Page 10; Line 11)

- There review is very comprehensive, which can leave readers a bit overwhelmed. The manuscript would benefit from a succinct summary sentence at the end of each section, and maybe an additional textbox/table/graphic the provides a short overall summary.

Response: Thank you for the suggestion. We have made the requested revision in

Table 2 (Page 57) and also added a succinct summary sentence at the end of each section.

Section 3 “From the above studies, increased IL-6 levels whether by stress, LPS or direct IL-6 injection produced depressive-like behaviors in rodents. The IL-6 effect in depression is further supported by Il6-deficient mice, which were resistant to the development of a depression-like behaviors following exposure to stress.” (Page 7; Line 6)

Section 4 “The above clinical studies suggested that higher blood IL-6 levels are associated with MDD or depression that manifests in patients with a physical disorder. IL-6 levels are also related to the development, manifestations and severity of MDD. However, evidence for the involvement of IL6 in the MDD susceptibility is currently inconsistent.” (Page 17; Line 16)

Section 5 “As shown above, there are a number of pharmacological agents and non-medication treatments for MDD that have shown effect on the IL-6 levels. IL-6 may represent a valid and highly effective target for the development of novel antidepressant treatments.” (Page 23; Line 20)

- On p. 21, psychological interventions are pictured as additional treatments that are used only if medication is not effective. This is misleading, as the treatment of depression always relies on many different interventions in parallel.

Response: We thank this reviewer for this comment. In the revised manuscript, we have modified this sentence.

“Psychological interventions (such as cognitive behavior therapy, mindfulness-based cognitive therapy, cognitive behavioral analysis system of psychotherapy, and interpersonal psychotherapy) is an effective treatment for MDD.” (Page 23; Line 6)

- There are two secions on genetics (p. 13 and p. 24). For consistency, these should be combined. Importantly, the authors should also briefly mention whether IL-6 variants are among the hits in large GWAS for depression (e.g. PMID:29700475).

Response: We thank this reviewer for this comment. In the revised manuscript, we combined the two sections together and added the report (PMID:29700475).

Section 4.5 (Page 14; Line 22)

“In a large genome-wide association analysis for MDD, 44 independent and significant loci were identified but IL6 variants were not among the hits [97].” (Page 16; Line 16)

- Animal studies of antidepressant medication are now included with human studies. For consistency, there should be a separate section (such as section 3.).

Response: We appreciate the insightful suggestion from the reviewer. We put this animal part into Sections 3 (Page 6; Line 16)

Minor

- The manuscript would benefit from being proof-read by a native English-speaker.

Response: As reviewer suggested, this revised manuscript has been edited through by a native English-speaking copyeditor to improve the English.

Many thanks!!

Reviewer 2 Report

Comments to the Authors:

This review provides an interesting association among IL-6 and major depressive disorder (MDD). It is well structured and covers basic studies, clinical results, and treatment suggestions. In particular, I appreciate that the authors mention subtypes of depression. However, there is a concern to be solved.
The authors do not mention the blood–brain barrier (BBB). Cytokines cross the BBB by regulated mechanism, not allowing substances like IL-6 to cross into the brain freely. I encourage authors to describe the BBB functions in a certain section. A paper by Capuron & Miller (Pharmacol Ther 2011;130:226-38) may be helpful. In addition, Tsuboi et al. (J Affect Disord 2019;249:385-93) hypothesized a pathway of IL-6 into depressive symptoms. Also, Maes et al. (Neuro Endocrinol Lett 2008;29:117-24) described the association between the BBB and MDD.

Minor points:

Page 22 Line 8-10
Are kynurenine hypothesis (authors mention it in Page 22 Line 9) and glutamate hypothesis also significant? Excitatory synapse hypothesis may be less important, though.

Page 24 Line 20-26
Add references. IL6 rs1800795 C allele increased the risk for depression in some cases but not in other cases.

Page 26 Line 20
Authors described add possible treatments; One or two sentences are necessary in the conclusion section.

Reference 92: HTML character codes appear.

Reference 97: ‘SSRI’ and ‘SNRI’ should be written with Capital letters.

Reference 97: ‘GSH’ should be written with Capital letters.

Author Response

This review provides an interesting association among IL-6 and major depressive disorder (MDD). It is well structured and covers basic studies, clinical results, and treatment suggestions. In particular, I appreciate that the authors mention subtypes of depression. However, there is a concern to be solved.

Response: We thank the reviewer for the thoughtful and constructive comments, which helped to improve our manuscript. We hope the reviewer finds that we have adequately addressed their concerns in the revised version of the manuscript.

The authors do not mention the blood–brain barrier (BBB). Cytokines cross the BBB by regulated mechanism, not allowing substances like IL-6 to cross into the brain freely. I encourage authors to describe the BBB functions in a certain section. A paper by Capuron & Miller (Pharmacol Ther 2011;130:226-38) may be helpful. In addition, Tsuboi et al. (J Affect Disord 2019;249:385-93) hypothesized a pathway of IL-6 into depressive symptoms. Also, Maes et al. (Neuro Endocrinol Lett 2008;29:117-24) described the association between the BBB and MDD.

Response: We thank the reviewer for the constructive suggestion. We added a section to mention the blood–brain barrier.

“The blood brain barrier (BBB) does not allow the relatively large cytokine molecules to freely pass through. However, as evidence suggests that neural, cellular and humoral pathways are used for cytokine signals to reach the brain [127]. The permeability of BBB may be potentially increased by circulating inflammatory factors, allowing access to the brain by pro-inflammatory cytokines released by activated macrophages and monocytes, such as IL-6 via the leaky regions of the BBB, including the circumventricular organs and the choroid plexus [127,128]. Moreover, there may be compromise of the intestinal barrier functions by activation of the inflammatory response system via an increase in IL-6 production [129]. Loss of function of the protective barrier may be caused by the latter, leading to the increase in the gram-negative bacteria translocation (leaky gut) [129].” (Page 24; Line 14)

Minor points:

Page 22 Line 8-10

Are kynurenine hypothesis (authors mention it in Page 22 Line 9) and glutamate hypothesis also significant? Excitatory synapse hypothesis may be less important, though.

Response: Thank you for pointing out this. We have revised this part as below.

“Studies had indicated that IL-6 may increase the activity of indoleamine-2,3-dioxygenase (IDO), which is responsible for the catalysis of tryptophan, leading to kynurenine pathway activation and decreasing central serotonin availability [135]. This results in the synthesis of the neurotoxic N-methyl-d-aspartate glutamate agonist quinolinic acid and 3-hydroxykynurenine thereby enhancing oxidative stress and contributes to neurodegeneration which characterise MDD particularly in late life [136].” (Page 25; Line 17)

Page 24 Line 20-26

Add references. IL6 rs1800795 C allele increased the risk for depression in some cases but not in other cases.

Response: Thank you for the suggestion. We have added the reference.

“Among the common functional IL6 genetic variants, some studies demonstrated that the IL6 rs1800795 C allele increased the risk for depression, however not by all [90,92-94,96].” (Page 17; Line 4)

Page 26 Line 20

Authors described add possible treatments; One or two sentences are necessary in the conclusion section.

Response: Thank you for the suggestion. We have added the below in the conclusion.

“While the need for novel antidepression therapies remains high, it is imperative that a greater understanding of the complex and apparently dichotomous effects of IL-6 manipulation in the brain is first acquired.” (Page 30; Line 19)

Reference 92: HTML character codes appear.

Response: We have corrected this. (new ref. number: 94) Page 45, Line 17

Reference 97: ‘SSRI’ and ‘SNRI’ should be written with Capital letters.

Response: We have corrected this. (new ref. number: 102) Page 46, Line 6

Reference 97: ‘GSH’ should be written with Capital letters.

Response: We have corrected this. (new ref. number: 104) Page 46, Line 14

Many thanks!!